# Inorganic Carbon Acquisition and Photosynthetic Metabolism in Marine Photoautotrophs: A Summary [note 1]

**DOI:** 10.3390/plants14060904

**Published:** 2025-03-13

**Authors:** Sven Beer, John Beardall

**Affiliations:** 1School of Plant Sciences and Food Security, Tel Aviv University, Tel Aviv 69978, Israel; 2School of Biological Sciences, Monash University, Clayton, VIC 3800, Australia; john.beardall@monash.edu

**Keywords:** algae, bicarbonate (HCO_3_^−^), CO_2_-concentrating mechanisms (CCMs), cyanobacteria, inorganic carbon (Ci), marine, photoautotrophs, seagrasses

## Abstract

The diffusive availability of CO_2_ for photosynthesis is orders of magnitude lower in water than in air. This, and the low affinity of ribulose-1,5-bisphosphate carboxylase/oxygenase (Rubisco) for CO_2_, implies that most marine photoautotrophs (cyanobacteria, microalgae, macroalgae and marine angiosperms or seagrasses) would be severely restricted were they to rely only on dissolved CO_2_ for their photosynthetic performance. On the other hand, the ~120 times higher concentration of bicarbonate (HCO_3_^−^) makes this inorganic carbon (Ci) form more available for utilisation by marine photosynthesisers. The most common way in marine macrophytes to utilise HCO_3_^−^ is to convert it to CO_2_ within acidic micro-zones of diffusion boundary layers (DBLs), including the cell walls, as catalysed by an outwardly acting carbonic anhydrase (CA). This would then generate an intra-chloroplastic (or for cyanobacteria intra-carboxysomal) CO_2_-concentrating mechanism (CCM). Some algae (e.g., the common macroalgae *Ulva* spp.) and most cyanobacteria and microalgae feature direct HCO_3_^−^ uptake as the most efficient CCM, while others (e.g., some red algae growing under low-light conditions) may rely on CO_2_ diffusion only. We will in this contribution summarise our current understanding of photosynthetic carbon assimilation of submerged marine photoautotrophs, and in particular how their ‘biophysical’ CCMs differ from the ‘biochemical’ CCMs of terrestrial C_4_ and Crassulacean Acid Metabolism (CAM) plants (for which there is very limited evidence in cyanobacteria, algae and seagrasses).

## 1. Introduction

Although oceans have supported oxygenic photosynthesis of cyanobacteria for close to 3.5 billion years, and about half of Earth’s primary (photosynthetic) production today occurs in our oceans [1], the study of the mechanisms involved in marine photosynthesis lags far behind that of terrestrial plants, which colonised land ‘only’ some 400 million years ago. This is despite the fact that the discovery and elucidation of the (almost) universal pathway of CO_2_ fixation and reduction, the Calvin–Benson–Bassham cycle (or the photosynthetic carbon reduction cycle, PCRC), was based on work using an, albeit freshwater, unicellular alga.

Two distinct differences between photosynthesis in terrestrial and aquatic environments are that (1) the diffusivity of solutes, including CO_2_, is orders of magnitude lower in water than in air and (2) irradiance is exponentially attenuated with depth such that the euphotic zone often extends only a few tens of metres down along a depth gradient [2]. Regarding photosynthetically active radiation (PAR), the dogma has been that net photosynthesis (photosynthetic activity corrected for mitochondrial respiration and other, minor carbon losses) on a diel cycle is positive (and thus supports growth) down to a depth where 1% of surface irradiance remains. However, it is now apparent that there are many exceptions to this, where net photoautotrophic growth occurs at much lower light levels [3,4,5,6]. Marine angiosperms (seagrasses, which colonised the seas secondarily only less than 100 million years ago) must support an often-extensive rhizome and root biomass and, thus, need up to 10% of surface light in order to grow [7,8,9]. While adaptations to various environments, including the various and varying irradiances, are a must in both terrestrial and marine ([1,2,10,11,12] and Chapter 9 in [13]) photoautotrophs, we will summarise here our current understanding of what we see as the main difference between the two groups, i.e., how the marine species in general, as well as using specific examples, cope with the scarce supply of CO_2_ in seawater while also featuring a generally low CO_2_ affinity of their ultimate carboxylating enzyme ribulose-1,5-bisphosphate carboxylase/oxygenase (Rubisco).

## 2. Marine Inorganic Carbon Sources for Carbon Assimilation

When in equilibrium with today’s atmospheric CO_2_ concentration of ~410 ppm (or 18 μM), and rising, the marine dissolved CO_2_ concentration at 20 °C and a salinity of 35 is some 25% lower (a value that depends on temperature and salinity). In addition, the diffusivity of solutes (including CO_2_) in water is orders of magnitude slower than in air. Therefore, and given the generally low CO_2_ affinity of Rubisco in marine photoautotrophs (see below), photosynthetic rates of the latter would be severely limited were they to depend on dissolved CO_2_ only. However, atmospheric CO_2_ (CO_2air_) reacts with water molecules to form carbonic acid (H_2_CO_3_), which dissociates to form bicarbonate (HCO_3_^−^) and carbonate (CO_3_^2−^) ions, according to Equation (1). These reactions are important as they define the relative concentrations of the various inorganic carbon species in an aqueous system such as seawater, and whereas CO_2_ can enter cells easily by diffusion, the charged bicarbonate ion cannot cross biological membranes without the use of transport proteins.
   CO_2air_ ↔ CO_2_ + H_2_O ↔ H_2_CO_3_ ↔ HCO_3_^−^ + H+ ↔ CO_3_^2−^ + H^+^(1)
18 μM |  14 μM     **CA** < 1 μM  ~1700 μM     ~300 μM
 Air    |   Seawater (at Ph 8.1, 20 °C and a salinity of 35)

As can be seen, the ionic concentrations in equilibrium with dissolved CO_2_ are strongly dependent on pH (H^+^), and for most submerged marine photoautotrophs, the most available Ci form (in terms of concentration) is HCO_3_^−^, which in today’s air-equilibrated seawater pH of 8.1 is ~120 times higher than that of CO_2_ (CO_3_^2−^ is not used in marine photosynthesis). Secondly, the hydration of CO_2_ to form H_2_CO_3_ and vice versa is a slow process with half-times approaching a minute. However, it becomes virtually instantaneous when catalysed by the enzyme carbonic anhydrase (CA), and the importance of this enzyme in marine photosynthetic Ci acquisition will be described below.

## 3. Assimilation of CO_2_ via Rubisco

The central enzyme of carbon assimilation in cyanobacteria, eukaryotic algae, marine (seagrasses) and terrestrial vascular plants is Rubisco, which is the first step in the PCRC (Equation (2)) for net assimilation of inorganic carbon to organic matter. Although other pathways for the conversion of CO_2_ to organic matter exist, those are found in non-oxygenic autotrophic bacteria [14]. The reactions of the PCRC and differences in its regulation among cyanobacteria and eukaryotic algae have been recently discussed in [15].

Rubisco will catalyse both the carboxylation of ribulose bisphosphate, to yield two molecules of phosphoglycerate, and its oxygenation to yield one phosphoglycolate plus one phosphoglycerate according to Equations (2) and (3) belowCO_2_ + H_2_O + D-ribulose-1,5-bisphosphate → 2 × 3-phosphoglycerate(2)O_2_ + D-ribulose-1,5-bisphosphate→3-phosphoglycerat + 2-phosphoglycolate(3)

The phosphoglycolate produced by the reaction shown in Equation (3) can be processed by the reactions of photorespiration, which sees the loss of one of the carbons as CO_2_, with the other carbon being recouped as glycerate. It has been shown [16] that, at least in the cyanobacterium *Synechococcus*, this is achieved through three pathways, i.e., (1) a ‘conventional’ photosynthetic carbon oxidation cycle (PCOC) involving glycolate dehydrogenase, and regeneration of glycerate via glyoxylate, glycine, serine and hydroxypyruvate, (2) the glycerate via tartronic semialdehyde or (3) conversion of glyoxylate via oxalate and formate to CO_2_. Further details of photorespiration in aquatic phototrophs can be found in [17].

Since the two reactions catalysed by Rubisco are competitive, using the same active site on the enzyme, the relative rate of the two reactions in a photosynthesising cell can be given by the specificity factor S_rel_ according to Equation (4),S*_rel_* = [K_0.5_(O_2_)•k_cat_(CO_2_)]/[K_0.5_(CO_2_)•K_cat_(O_2_)](4)
where k_cat_(CO_2_) = CO_2_-saturated specific rate of carboxylase activity of Rubisco (mol CO_2_ mol^−1^ active sites s^−1^), K_0.5_(CO_2_) = concentration of CO_2_ at which the CO_2_ fixation rate is half of k_cat_(CO_2_), k_cat_(O_2_) = O_2_-saturated specific rate of oxygenase activity of Rubisco (mol O_2_ mol^−1^ active site s^−1^) and K_0.5_(O_2_) = concentration of O_2_ at which the O_2_ fixation rate is half of k_cat_(O_2_).

The lower the selectivity factor, the higher will be the impact of oxygenation on net photosynthetic CO_2_ fixation and photorespiration. For organisms with the lowest S_rel_ values, the kinetics of Rubisco do not allow for net fixation of CO_2_ in air-equilibrated seawater (or freshwater), irrespective of the pathway of glycolate metabolism used in photorespiration [17]. In terrestrial C_3_ plants, K_0.5_(CO_2_) values are generally lower than those of the algae and cyanobacteria. C_4_ terrestrial plants have values higher than those of C_3_ plants, though the values are closer to air-equilibrium CO_2_ levels than in algae and cyanobacteria [18]. In the following, we will compare the affinities of different Rubiscos for CO_2_ with those of the whole organism, leading to the conclusion that most marine oxygenic photoautotrophs need a CCM to carry out significant rates of net photosynthesis.

## 4. Kinetics of Rubisco vs. Whole-Cell Photosynthesis

### 4.1. Cyanobacteria and Microalgae

The forms of Rubisco vary considerably across the cyanobacterial and microalgal groups and show different kinetic properties and selectivity toward CO_2_ and O_2_. The general trend across all photoautotrophs, though, is that a low K_0.5_(CO_2_) and a high S_rel_ are correlated with a low k_cat_(CO_2_) and vice versa. Cyanobacteria tend to possess Rubiscos with high K_0.5_(CO_2_) and somewhat lower selectivity factors than the eukaryotic microalgae, which may reflect the cyanobacterial evolutionary origins when CO_2_ levels were higher and O_2_ concentrations lower than the present-day values [19,20,21]. Affinities of Rubiscos from eukaryotic algae tend to be higher (lower K_0.5_(CO_2_)) and S_rel_ values higher. Values of the kinetic properties of Rubiscos from a range of cyanobacteria and microalgae are given in Table 1. This table includes values from both marine and freshwater species, but differences in the values reflect more the taxonomic position of the species than their salinity environments.

**Table 1 plants-14-00904-t001:** Kinetic properties of Rubiscos from different cyanobacterial and microalgal groups. Note these data include values from freshwater as well as marine species.

Organisms	Rubisco	K_0.5_ CO_2_ (μM)	S_rel_ (mol mol^−1^)	K_cat_ (mol CO_2_ mol^−1^ Active Sites s^−1^)	Reference
β-cyanobacteria	Form IBc	200–260 *	35–56	2.6–11.4	[22] and references therein
*Prochlorococcus marinus*	Form 1Ac	750		4.7	[23]
(α-cyanobacterium)		309	60	6.6	[24]
Green algae	Form 1B	29–38	61–83		[22] and references therein
Diatoms	Form 1D	23–68	57–116	2.1–3.7	[18] and references therein, [25]
Synurophyceae	Form 1D	18.2–41.8			[26] [27]
Olisthodiscophyceae	Form ID				
*Olisthodiscus*		59	100.5	0.83	[28]
Coccolithophorids	Form ID				
*Pleurochrysis carterae*		17.7	102	3.3	[29]
*Emiliania huxleyi*		72–200			[30,31]
Dinoflagellates ^#^	Form II				
*Amphidinium carterae*			~37		[22]
Rhodophyta	Form 1D				
*Porphyridium purpureum*		22	129	2.6	[28]
*Cyanidium*		6.6–6.7	224–238	1.3–1.6	[32]

* A minority of reports have lower values down to 105 μM. ^#^ Kinetic data on dinoflagellate Rubiscos are limited due to the instability of their Form II Rubiscos.

We note that the values of K_0.5_(CO_2_) for Rubisco are in most cases higher than the air-equilibrium CO_2_ concentrations generally found in various aquatic environments (10–25 μM, depending on temperature, salinity and the presence and density of various respiring and photosynthesising organisms, see Section 7). Consequently, air-equilibrium levels of CO_2_ would, all other things being equal, be limiting to carbon assimilation by cyanobacteria and microalgae. Furthermore, most cyanobacteria and microalgae (including picophytoplankton [33]) show much higher affinity for Ci in whole-cell photosynthesis than is exhibited by the isolated Rubisco. To cite a few examples, values for K_0.5_(CO_2_) in Ci-dependent photosynthesis of cells of *Phaeodactylum tricornutum* have been reported as 0.53–4.5 μM [34,35] whereas those for its Rubisco are 28–41 μM (see [36] and references therein). Similarly, while [37] report that *Chaetoceros muelleri* cells have K_0.5_(CO_2_) values of ~0.26 μM in photosynthesis, [25] quote an equivalent value for Rubisco from this species of 23 μM. The marine cyanobacterium *Synechococcus* PCC7002 with Form 1Bc Rubisco has a K_0.5_(CO_2_) for Rubisco of 185 μM [36] and the equivalent value for photosynthesis of ~0.7 μM [38]. There are only a few exceptions to this trend, with the most rigorous examples coming from freshwater Synurophyceae, which have evolved Rubiscos with a high affinity for CO_2_ ranging from 18 to 42 μM [26] compared to similar values of whole-cell photosynthesis of 22–45 μM.

It is clear that cyanobacteria and most microalgae have evolved mechanisms to enhance CO_2_ concentrations at the active site of Rubisco such that carbon assimilation is at, or close to, CO_2_ saturation at air-equilibrium levels in the surrounding medium. These CO_2_-concentrating mechanisms (CCMs) are discussed below in Section 5.

### 4.2. Macroalgae and Seagrasses

Reported K_0.5_(CO_2_) values for macroalgae of all three phyla range from ca. 20 to 85 μM ([18,20,39,40,41,42,43] and references therein). Lower-end values (10–20 μM) were reported for several red algae (and especially for *Griffithsia monilis* (<10 μM)) as compared to green and brown taxa [18], but some brown algae also show low values (e.g., *Laminaria hyperborea* [39]). As a general trend, however, we agree with the conclusion of [18] that the red algal Rubiscos feature lower K_0.5_(CO_2_) values than the other two phyla. This higher affinity for CO_2_ in red algae may correlate with many of their species not possessing CCMs but relying on diffusive entry of CO_2_ for their photosynthetic needs (see Section 5.2 below). We also summarise here that most green and brown macroalgae feature K_0.5_(CO_2_) values that are significantly higher than the air-equilibrated marine CO_2_ concentration and, thus, are in need of CCMs (the mechanisms of which will be detailed in Section 5.2).

While the different K_0.5_(CO_2_) values can partly be due to the red and brown macroalgae possessing the Form 1D while the green algae and seagrasses contain Form 1B Rubiscos, the different values for species that have the same form are less understood. However, the link between the evolutionary occurrence of various Rubiscos and the evolution of marine photoautotrophs has been reviewed in [43] and has gained much recent interest [20,36,44].

Seagrasses are marine angiosperms that colonised shallow softbottom areas some 100 million years ago. Like their terrestrial counterparts from which they evolved, they possess roots and rhizomes as well as flowers and pollination (although meadows mostly grow clonally by rhizome extensions). The photosynthetic traits of these plants have been reviewed in [45,46,47]. Values of K_0.5_(CO_2_) for Rubiscos in seagrasses have recently been measured to be around 40 μM, with the values being significantly lower in the absence of O_2_, indicating the presence of photorespiration. These include the Mediterranean species *Posidonia oceanica* (K_0.5_ (CO_2_) ~45 μM), Mediterranean *Cymodocea nodosa* (37 μM) and the temperate *Zostera marina* (32 μM) under air-equilibrium O_2_ concentrations. As is the case for macroalgae, these plants must thus also possess CCMs.

We conclude in this section that the low availability of CO_2_ in seawater and the low affinity of Rubiscos for CO_2_ necessitate the presence of a CCM in almost all marine photoautotrophs. As follows, we will see that such CCMs are most often based on the use of HCO_3_^−^, which, again, is present at a much higher concentration in seawater than CO_2_.

## 5. Inorganic Carbon Acquisition and ‘Biophysical’ CCMs

Measurements of the kinetics of whole-cell photosynthesis vs. CO_2_ in most marine photoautotrophs show K_0.5_(CO_2_) values that are much lower than for their isolated Rubiscos. This higher affinity of intact cells/tissues implies that the CO_2_ concentration at the active site of Rubisco is higher than in the bulk medium, i.e., a CCM is present. This is supported by direct measurements of internal CO_2_ pools with CO_2in_ vs. CO_2out_ ratios ranging from ~2 fold in symbiotic dinoflagellates, 20-fold in the macroalga *Ulva fasciata*, to 800–900 in cyanobacteria, though there is considerable variation depending on species and environmental conditions (see Table 1 in [17] and references therein). There is also other, indirect evidence for CCM activity including carbon isotope discrimination values against ^13^C being much less negative than for isolated Rubisco, low or absent inhibition of photosynthesis by O_2_ and low CO_2_ compensation points in photosynthesis vs. Ci measurements.

Inorganic carbon assimilation by marine photoautotrophs is, in most cases, driven by the use of HCO_3_^−^, the main Ci source in seawater. Bicarbonate acquisition can be demonstrated in several ways. One approach, which may also be the easiest one, is to measure the final pH values in pH-drift experiments [48]. In short, cell suspensions, macroalgae or seagrasses (or parts thereof) are inserted into a closed illuminated chamber in which the rise in pH is recorded as they photosynthesise [49]. If the final pH is around 9 or higher, then this suggests HCO_3_^−^ use since the CO_2_ concentration is extremely low at the high pH values generated in closed systems. If, on the other hand, the final pH is 8.5 or less, then HCO_3_^−^ is not used since this ion is present but obviously does not contribute substantially to raise the pH through photosynthesis. Thus, those (albeit few) algae and plants would be CO_2_-only users. Burns and Beardall [34] used the pH dependence of K_0.5_(CO_2_) and independence of K_0.5_(HCO_3_^−^) to infer HCO_3_^−^ use in a number of marine microalgae. It should be noted that this approach may not yield definitive proof of one Ci form being used exclusively, since a combination of the two Ci forms may be utilised as the pH changes in the system. Another way to demonstrate HCO_3_^−^ use is to determine the ^13^C vs. ^12^C contents in algal and plant material vs. those of the Ci sources in the seawater. These values are expressed as Δ^13^C (δ^13^C if the source ^13^C:^12^C ratio is not determined). Isolated Rubiscos show values close to −27‰ to −30‰ but HCO_3_^−^ use and activity of a CCM is reflected in less negative values. A summary of the use of δ^13^C in investigations of marine photoautotrophs may be found in [50]. Isotope disequilibrium techniques as described by [51,52] can also be used to indicate HCO_3_^−^ vs. CO_2_ use. Alternatively, membrane inlet mass spectroscopy as described for the marine *Synechococcus* [38] can be used to directly determine the rates of uptake of CO_2_ and/or HCO_3_^−^.

### 5.1. Cyanobacteria and Microalgae

Use of HCO_3_^−^ and its role in driving CCM activity has been demonstrated in a range of cyanobacteria and microalgae, and CO_2_ can also cross the plasma membrane by diffusion, in some cases aided by aquaporins [15,17,53].

In cyanobacteria, there are multiple systems for active Ci entry into cells and accumulation of CO_2_ within the carboxysome [54], with some differences in the transporters between marine and freshwater species [55]. Most of these transport systems involve direct active transport of HCO_3_^−^ or an energised conversion of CO_2_ to HCO_3_^−^ via a NAD(P)H dehydrogenase, which effectively acts like a Ci pump, even though direct active transport of CO_2_ has not occurred ([53] and references therein, [55]). Enhanced HCO_3_^−^ supply to the carboxysome (the site of Rubisco activity in cyanobacteria) leads to increased CO_2_ levels, involving the activity of a carboxysomal CA (see [55] and references therein).

The CCMs of eukaryotic algae have been investigated in detail for the model freshwater green alga *Chlamydomonas* [56], but those in marine microalgae are less well elucidated. Nonetheless, HCO_3_^−^ transporters (SCL4-type transporters) associated with the plasma membrane have been identified in the marine diatoms *Phaeodactylum tricornutum* and *Thalassiosira pseudonana* [57]. Energy for this process appears to be associated with linear electron flow involving both photosystems II and I [58]; in eustigmatophyte algae, respiratory-derived ATP is used [59]. There is also evidence, based on observations for some species, of active transport of CO_2_ [60] and HCO_3_^−^ and CO_2_ accumulation by isolated chloroplasts [61] and there is evidence for SCL4-type transporters at the chloroplast envelope of diatoms [57,58], and a range of transporters of inorganic carbon have been postulated for all four of the plastid envelope membranes [62]. Ultimately, CO_2_ is accumulated at the active site of Rubisco—in many algae, this is in the pyrenoid within the chloroplast. As with carboxysomes, the localisation of Rubisco, tightly packed in a microcompartment, permits inorganic carbon accumulation with limited leakage.

In eukaryotic microalgae (but not cyanobacteria), Ci acquisition can be modulated by an extracellular CA associated with the cell wall. This converts HCO_3_^−^ to CO_2_ in the periplasmic space, thereby increasing CO_2_ concentrations locally and assisting its diffusion across the cell wall and plasma membrane. A range of other CAs are present in various compartments within the eukaryotic algal cell and are involved in maintaining the equilibrium between CO_2_ and HCO_3_^−^ [63].

### 5.2. Macroalgae and Seagrasses

The macroalgal photosynthetic traits studied the most are for various species of *Ulva* (reviewed in [64]), and this genus will therefore be used here as an example applicable for many others that have been studied, as well as what will most probably be found in future investigations. So, among other genera of red, brown and green algae, *Ulva* spp. (including what was once named *Enteromorpha*) stand out as being extremely efficient HCO_3_^−^ users, i.e., a macroalga in which additional CO_2_ or Ci beyond the normal seawater Ci composition will not significantly increase photosynthetic rates ([65,66] and Figure 7.5 in [13]). Species of this genus can use HCO_3_^−^ in two different ways (see below) and the Rubisco K_0.5_(CO_2_) for subtropical *Ulva fasciata* was found to be 70 μM [67]. In comparison with today’s air-equilibrated CO_2_ concentration of ~14 μM, at the salinity and temperature used, it can be ascertained that a diffusive CO_2_ equilibrium could yield only ~5% of this alga’s photosynthetic potential as based on its Rubisco carboxylation activity—if assuming that the CO_2_ concentration inside the photosynthesising cells would be in equilibrium with that of the seawater (which is an overestimation). Thus, like most macroalgae, *Ulva* must possess a CCM.

A spin-off from pH-drift experiments is to actively vary the pH with HCl or NaOH within an illuminated closed system containing algae or seagrasses (or, usually, parts thereof) while measuring both the pH and photosynthetic rates, the latter usually by O_2_ evolution. Given the total Ci in natural seawater (~2.2 mM) and the pH, the Ci composition can then be calculated. The results of experiments using such systems showed already early on that the common macroalgal genus *Ulva* used HCO_3_^−^ from seawater in such an efficient way that this Ci form saturated photosynthesis without any need for supplemental CO_2_ ([66,68] and Figure 7.5 in [13]). This was not the case for seagrasses [69,70] where both subtropical [71,72] and, later, temperate and Mediterranean [73,74] species could use HCO_3_^−^ under normal seawater Ci conditions, but increased their photosynthetic rates, often more than doubling them, as CO_2_ was released by lowering the pH to <8 (see Figure 7.12 in [13]).

Our conclusion of the above-mentioned approaches is that marine macrophytes (macroalgae and seagrasses) in general use HCO_3_^−^ as their external Ci source for photosynthesis, but that seagrasses do so less efficiently and, thus, are Ci-limited in today’s air-equilibrated seawater and, accordingly, may benefit from future increases in atmospheric and, consequently, dissolved CO_2_ levels (see, however, Section 8). This general conclusion was somewhat challenged recently [20] as being an underestimation of the capacity of seagrasses to photosynthesise at close to the HCO_3_^−^ concentration of today’s oceans (see also Section 7).

Two cautionary notes before continuing as follows: Most experiments on photosynthetic properties of marine photoautotrophs are performed in laboratories. For the macrophytes, this usually means that parts of thalli or seagrass leaves are cut into sections so as to fit into small O_2_ electrode chambers. Such (mis)handlings may show results that are different from those measured in situ. This was shown for the shallow-growing tropical seagrasses *Halophila ovalis* and *Cymodocea serrulata*, in which the results of previous laboratory experiments showed that they were Ci-limited in the ~2.2 mM-Ci normal seawater [72,75]. However, when the measurements were performed in situ on intact plants, both species were Ci-saturated [76]. Secondly, buffers have often been used in order to keep a certain pH while measuring O_2_ evolution. However, it was shown that TRIS buffer itself lowered photosynthetic rates even at normal seawater pH and Ci conditions [77]. This led to the conclusion that, e.g., the common temperate seagrass *Zostera marina* was highly dependent on active trans-membrane pumps producing protons (H^+^) toward the diffusion boundary layer (DBL, including the cell walls), which, when using buffers, were neutralised.

As mentioned before, the most common way for marine photoautotrophs to utilise HCO_3_^−^ is to convert it to CO_2_ within their DBL as catalysed by membrane-bound, extracellularly acting CA (Figure 2 in [64]). If the pH within the DBL where CA activity is present is lower than that of the surrounding seawater (e.g., in acid zones [47]), then this will account for an efficient conversion of HCO_3_^−^ to high concentrations of CO_2_ that can diffuse through the plasma membrane into the photosynthesising cells. This mechanism of Ci utilisation in macroalgae and, especially, in most seagrasses, is today common knowledge that has been detailed in, e.g., Chapter 7 in [13] and [20,45,46,47,78]. The easiest way to detect this form of HCO_3_^−^ use is to add the membrane-impermeable CA inhibitor acetazolamide (AZ) while measuring photosynthesis; if rates cease, then this is a strong indication of this mechanism being used. If not, then either another way to use HCO_3_^−^ is in effect or CO_2_ only is used (see the following section).

The most efficient “other way” to use HCO_3_^−^ from seawater is by its direct uptake (Figure 3 in [64]). Being an ion, HCO_3_^−^ cannot easily diffuse through plasma membranes and, so, needs to be transported. For subtropical *Ulva* sp., it was found that photosynthesis was fully inhibited by 4,4′-diisothiocyanostilbene-2,2′disulphonate (DIDS) [79], which, till then, had been used as a classic inhibitor of HCO_3_^−^ transport in red blood cells (RBCs) via a membrane-bound anion exchange (AE) protein. It was further found that the AE from *Ulva* and from RBCs shared very similar properties [79,80] with the main difference that RBCs exchanged HCO_3_^−^ for chloride (Cl^−^) while *Ulva* exchanged it for hydroxyl ions (OH^−^) [81]. Interestingly, when AZ-sensitive temperate *Ulva lactuca* was exposed to pH 9.8 for some 10 h, it converted to the state of direct, DIDS-sensitive HCO_3_^−^ uptake alga similar to those *Ulva* spp. growing in the subtropical waters of the Mediterranean [82]. When in this state, *Ulva* had a 10 times higher affinity for HCO_3_^−^ than the one growing in temperate waters. It was concluded that, ultimately, the pH near the plasma membrane was the trigger for the alternative ways in which subtropical *Ulva fasciata* and temperate *Ulva lactuca* used HCO_3_^−^. The addition of DIDS was subsequently used by several researchers as a way to indicate direct uptake of HCO_3_^−^, so, besides *Ulva*, *Chaetomorpha melagonium* [83], *Macrocystis pyrifera* [84] and *Posidonia oceanica* [85] could also transport HCO_3_^−^ into their photosynthesising cells.

There is, of course, an energy cost for CCMs [86], which obviously is met by the irradiances where CCM-requiring algae and seagrasses grow. At lower light (at depths) and in colder waters, some macroalgae may not need CCMs and can, like terrestrial C_3_ plants, use the, albeit slow, diffusional supply of CO_2_ for their lower photosynthetic rates. These algae include several rhodophytes [87], but they still represent a minority among macroalgae, which, again, mostly use HCO_3_^−^. The same goes for algae growing in subarctic or Arctic areas [88] where photosynthetic rates are restricted by low temperatures (see, however, a report of two polar red algae being HCO_3_^−^ users and possessing CCMs [89]).

Several parameters found in macroalgae strongly suggest that they (a) use HCO_3_^−^ as their external Ci source under today’s marine Ci composition and (b) that they possess CCMs. These include low CO_2_ compensation points for many macroalgae including the common species *Enteromorpha* (now *Ulva*) *intestinalis*, *Ulva lactuca*, *Porphyra umbicalis*, *Palmaria* (then *Rhodymenia*) *palmata*), *Fucus serratus* and *Pelvetia canaliculata*; in all cases were the CO_2_ compensation concentrations well below ~10 μM [90], showing that virtually no CO_2_ is leaked outward even at low CO_2_ concentrations and, thus, indicating an efficient Ci utilisation system (a CCM) that keeps all CO_2_ within the photosynthesising cells to be used by photosynthesis. For most macroalgae, there is, however, a lack of direct evidence that intracellular CO_2_ concentrations surrounding Rubisco are higher than those of the surrounding seawater. As far as we know, such estimates are limited to, again, *Ulva fasciata* where intracellular Ci concentrations within the photosynthesising cells were found to be 220 μM [67]. With a Rubisco K_0.5_(CO_2_) of 70 μM, and assuming an intracellular pH of 7.2 [91] and that ~200 μM CO_2_ was present also within the chloroplasts, this meant that this alga could concentrate CO_2_ so as to fully saturate its Rubisco.

Seagrasses in general increase their rates of photosynthesis when CO_2_ is allowed to increase above air-equilibrated seawater concentrations (~14 μM today). This can be performed by lowering the pH or adding Ci above the natural seawater concentration in a closed system ([20,73,74,75,92], see Figure 7.12 in [13] for an early example). There are a whole range of affinities to Ci exhibited by various species; some seem to be severely limited in their HCO_3_^−^ utilisation capacity (e.g., *Thallasia testudinum*) [73] while others are more efficient (e.g., *Halodule wrightii*, *Syringodium filiforme* and *Posidonia oceanica* are less affected by additional Ci or CO_2_ [72,74,93]). All summaries on Ci utilisation and CCMs, specifically in seagrasses [20,45,46,47,94], generally agree on HCO_3_^−^ being the main external Ci form used in seagrass CCMs.

Rubisco originally evolved when CO_2_ levels were very much higher, and O_2_ levels lower, than in the present day; most probably during periods when CO_2_ fell from very high values to 2–16 times the present atmospheric level [19,20,21]. Adaptations of Rubisco’s capacities, and how it co-evolved with photoautotrophs, have been thoroughly reviewed recently [20] and will not be treated further here.

Some variations on the theme of external CA-mediated HCO_3_^−^ use in some macroalgae and seagrasses have also been reported. One originated in 2001 from experiments in which buffers (usually TRIS) were (mis)used for keeping certain pH values within O_2_ electrode systems. It was then found that the TRIS buffer itself at the natural seawater pH had a negative effect on the photosynthetic rates of the seagrass *Zostera marina* [77]. This led to the assumption that H^+^ release was an alternative way for this seagrass to utilise HCO_3_^−^ by co-transporting the two ions into the photosynthesising cells; the TRIS buffer would interfere by neutralising the H^+^ efflux [94] (see Figure 7b in [13]). Another way of using HCO_3_^−^ was also suggested for the Indo-Pacific seagrass *Halophila stipulacea* in which both TRIS and AZ together could impede CO_2_ formation and fluxes of CO_2_ into the cells (Figure 7.13c in [13]). This was also found for several other species such as *Halodule wrightii*, *Halophila ovalis* and *Cymodocea rotundata* [95]. However, most works point to the external CA-mediated conversion of HCO_3_^−^ to CO_2_ as the most common way of Ci acquisition in seagrasses (Figures 7.7 and 7.13a in [13]).

Photorespiration (the beginning stage of which is given in Equation (3)) can be present in marine photoautotrophs [45,90], especially when lacking efficient CCMs. This is true, for instance, in the seagrass *Zostera marina* and, often seen as a wasteful process, can lower photosynthetic rates significantly, especially under conditions of low Ci and high O_2_ levels such as generated when growing together with the highly efficient photosynthesiser *Ulva* ([96] and see Section 7). On the other hand, photorespiration can also protect the seagrass from photodamage by dissipating solar energy at high irradiances. The presence of photorespiration was indicated by a lower gross O_2_ evolution rate under natural O_2_ conditions than when O_2_ was reduced.

## 6. Alternative ‘Biochemical’ Modes of Inorganic Carbon Utilisation in Some Marine Photoautotrophs

The CCMs described above are sometimes referred to as ‘biophysical’ CCMs because they involve active or facilitated transport of Ci into the photosynthesising cells. Although uncommon, there have been reports of phosphoenolpyruvate carboxylase (PEPC) and phosphoenolpyruvate carboxykinase (PEPCK) being primary carboxylases present in micro- and macroalgae and leading to ‘biochemical’ CCMs similar to those found in terrestrial C_4_ and CAM plants.

### 6.1. Cyanobacteria and Microalgae

Despite early reports of C_4_-like photosynthesis in a marine diatom [97], later work [98] ascribed the observed labelling patterns and enzyme activities to high rates of anaplerotic β-carboxylation reactions, necessary to ‘top up’ intermediates of the TCA cycle. While single-cell C_4_ photosynthesis has been more recently reported by [99,100], it is now more generally accepted that among the microalgae, this is only present in the diatom *Thalassiosira weissflogii*, which shows C_3_-C_4_ intermediate Ci assimilation [17,101,102,103]. All other cyanobacteria and microalgae examined possess a standard form of the PCRC, though there are several phylogenetic variations in the regulation of the pathway (see [15] for a recent review).

### 6.2. Macroalgae and Seagrasses

According to early work, it was claimed that the common macroalga *Ulva* must be a C_4_ alga based on its low Ci compensation point and O_2_-insensitive photosynthesis, as well as its high activities of PEPC and PEPCK relative to Rubisco. However, that was postulated before ‘biophysical’ CCMs became known and accepted, and marine C_4_ macroalgae were later seen as rare exceptions (i.e., we do not know of any except *Udotea flabellum* [104,105]); rather, from the mid-1970s, ^14^C-pulse/^12^C-chase experiments showed that several macroalgal species including *Ulva lactuca* [106], another *Ulva* sp. [42], and what was then called *Enteromorpha* (now *Ulva*) *compressa* [107], featured typical C_3_ incorporation patterns. While some workers found high levels of PEPC and PEPCK in *Ulva prolifera* [108], they still agree that this species is basically of the C_3_ type. Another concern is the following: the genus *Ulva* has at least 100 separate species [109], but they are often hard to tell apart. In our experience, however, all those *Ulva* forms (or alleged species) we have investigated perform similarly in terms of photosynthetic characteristics, including the feature of switching between the external CA-mediated and the direct HCO_3_^−^ uptake mode. Low-level CAM has also occasionally been reported for members of the Fucales [110]; however, that too is a very rare exception [27].

Some macroalgae can photosynthesise and grow using diffusional CO_2_ acquisition only. These include some red algae [87,88] growing in low-light and low-temperature environments where, apparently, photosynthetic rates are so low as not to require a CCM. On the other hand, this would also alleviate the need for energy to drive a CCM. Interestingly, red algae as a whole do tend to have lower K_0.5_(CO_2_) values (i.e., higher affinities for CO_2_ (see Section 4.2 above)) than other algal phyla and, so, may be less in need of a CCM.

We conclude from parts 5 and 6 that marine photosynthetic organisms in general possess CCMs that are different from the ‘biochemical’ ones of terrestrial C_4_ and CAM plants and, rather, are ‘biophysical’ and mainly based on HCO_3_^−^ utilisation.

## 7. Inorganic Carbon Acquisition in Various Marine Environments

Active acquisition of Ci and CCM activity are downregulated by elevated CO_2_ supply where CO_2_ supply by diffusion is sufficient to meet the needs of growth. CCMs appear also to be downregulated under low light. This seems to be the case for a number of red macroalgae grown at low light [111] as well as for the freshwater cyanobacterium *Anabaena variabilis* [112] where light supply is insufficient to drive CCM activity. Energy constraints on CCM activity are also seen under conditions of P limitation (see [113] and references therein). In the marine diatom *Chaetoceros muelleri*, external CA and CCMs were up-regulated as the cell density in cultures increased to a point where uncatalysed rates of CO_2_ supply from HCO_3_^−^ were insufficient to meet the demands of carbon fixation [37]. It has been suggested that CCMs allow increased nutrient use efficiencies under nutrient-limiting conditions, but evidence for N limitation and Fe limitation is mixed, and a detailed discussion of environmental effects on carbon acquisition and CCM regulation is provided by [113,114].

Regarding macrophytes, we can give three examples of how photosynthetic Ci acquisition modes of the prolific algae *Ulva* spp. can influence other macrophytes. Firstly, the unique ability of *Ulva* spp. to switch between external CA-mediated HCO_3_^−^ use and the very efficient direct uptake of the ion (making the alga 10 times more efficient in HCO_3_^−^ use than when in the external CA mode, see Section 6) can be connected with the climates where they grow. For example, *Ulva fasciata* growing in the Eastern Mediterranean where conditions of high temperatures and high irradiances are conducive to high photosynthetic rates, its HCO_3_^−^ exchange with OH^−^ will generate high pH values in its DBL and in isolated surroundings, causing CO_2_ concentrations to approach zero. If so, epiphytic or other algae will be unable to grow in its surroundings. However, when this *Ulva* is transferred to the temperate, lower irradiance waters of the Swedish West Coast, it switches to the external CA mode [82]. Secondly, *Ulva intestinalis* can generate its own high-pH surroundings by HCO_3_^−^/OH^−^ exchange in isolated rockpools where the conditions during summertime are also conducive to high photosynthetic rates. Thus, the photosynthetic mode of this alga can effectively hinder other genera from the nearby open waters, which feature external CA-mediated Ci acquisition, to grow there [115]. Thirdly, it was shown that the generally higher photosynthetic rates of *Ulva lactuca* than of the seagrass *Zostera marina* could increase the pH and, so, lower the Ci and CO_2_ and increase the O_2_ concentrations to such values that they caused the seagrass to photorespire [96]. Similarly, it was shown that ulvoid algae could deplete seawater conditions such that it affected the subtropical seagrass *Thalassia hemprichii* [116]. We estimate that future research will illuminate more of these interactions based on photosynthetic capabilities.

## 8. Future Scenarios

Our planet is currently experiencing a period of environmental change, which, albeit more rapid than in the past, will gradually result in elevated sea surface temperatures, higher CO_2_ concentrations, lower ocean pH (ocean acidification, OA) and increased nutrient limitations due to enhanced stratification of water columns. All of these factors are likely to affect the physiological performance and population compositions of aquatic phototrophs. Consequences of these climate-induced changes for marine photosynthetic organisms have been reviewed in [117,118].

At first sight, it would be predicted that aquatic photoautotrophs without, or with a weak, CCM, and thus showing a low affinity for CO_2_ or HCO_3_^−^ use in Ci acquisition, might show improved performance in the higher CO_2_ environments of the future [65,119,120,121]. That said, such predictions have not always been borne out, at least in macroalgae, possibly due to the need to divert resources and energy to maintain cellular homeostasis under OA. Van der Loos et al. [122] analysed reports on OA effects on a range of micro- and macroalgae and showed that in most species, there was little or no influence of OA on performance, though in other cases, there were a range of effects from decreases in photosynthesis and growth to large stimulatory effects, and competing effects on, e.g., the *Macrocystis pyrifera*’s microbiome [123]. In the study of macroalgae by [122], of 55 macroalgae possessing CCMs, 21 showed no response to elevated CO_2_, 15 exhibited increased growth and 5 had decreased growth relative to that in ambient seawater. Of the only five non-CCM species that were tested, three showed unaffected growth rates but enhanced growth in the other two species. However, it is clear that the effects on growth are highly variable across taxa and not always predictable from the relationship between photosynthesis and Ci [124]. Furthermore, few studies have considered evolutionary changes in populations rather than short-term acclimation responses in relation to global change and those that have been carried out have mostly involved microalgae [117,125,126]. Evolutionary studies on macroalgae and seagrasses would thus be an important area for future research.

## 9. Summary

Marine photoautotrophs in general use the 120 times higher HCO_3_^−^ than CO_2_ concentration in seawater for their photosynthetic needs;There are several ways in which cyanobacteria and microalgae can acquire Ci from seawater, including diffusion or active transport of CO_2_. For many microalgae, however, as well as for macroalgae and seagrasses, the most common way is to convert HCO_3_^−^ to CO_2_ via membrane-bound CA activity associated with the periplasmic space. Another, more efficient way to acquire HCO_3_^−^ is by its direct uptake, mediated, at least in the macroalga *Ulva*, by an anion exchange protein bound to the plasma membrane;Because marine photoautotrophs contain Rubiscos with lower affinities for CO_2_ than terrestrial C_3_ plants, and given the slow diffusional supply of this Ci form in seawater, they are in need of (and typically possess) CCMs in order to partly or fully (depending on species and surrounding conditions) saturate Rubisco with CO_2_ so as to optimise photosynthetic and growth rates. Some (mainly red) algae, however, can under low irradiance utilise only CO_2_ by diffusion;The ‘biophysical’ CCMs of marine photoautotrophs are different from the ‘biochemical’ CCMs of terrestrial C_4_ and CAM plants as they rely on extracellular HCO_3_^−^ supplying CO_2_ to their Rubiscos;Photoautotrophs using C_4_ and CAM pathways for inorganic carbon fixation are very rare in marine environments, but C_4_ metabolism may in some cases have an anaplerotic carboxylation role;While many macroalgae and all seagrasses investigated in laboratory conditions require additional CO_2_ to fully saturate carbon fixation, their performance in situ may be different such that they are closer to CO_2_ saturation without additional CO_2_ or Ci;Responses to future changes in CO_2_ levels would appear to be very species-dependent and also influenced by the modulation of CCM activity by other environmental conditions such as light and nutrient levels.

## Data Availability

Data is contained within the article.

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
