# Peer review of "Inorganic Carbon Acquisition and Photosynthetic Metabolism in Marine Photoautotrophs: A Summary"

_plants, 2025, doi:10.3390/plants14060904_

Round 1

Reviewer 1 Report

Comments and Suggestions for Authors

The manuscript titled "Inorganic Carbon Acquisition and Photosynthetic Metabolism in Marine Photoautotrophs: A Summary" presents a variety of mechanisms for carbon acquisition and, especially, carbon concentrating mechanisms in oxygenic phototrophs. The review presents a range of up-to-date information regarding the diversity of these metabolisms, primarily in algae, most notably Ulva species. While the information is generally straightforward, there are some organization and writing issues that impede the narrative, most of which are noted in the comments below.

The work is valuable and informative, though it is a bit focused on a narrow range of phototrophs. Much less information is provided for, for example, cyanobacteria, picophytoplankton, and other major marine taxa. CCM discussions, which constitute a large component of the work, don't cover the role of carboxysomes or other such "storage" solutions.

Minor comments and notes:

Lots of CCM talk before its section (5). Would it be worth rearranging to make this more logical?

72    Potential typo or odd usage at "which e.g."
76    Odd and weirdly on the nose wording in "and the importance of this enzyme in marine photosynthetic Ci acquisition will become apparent below." The "importance" doesn't "become apparent below" in the text, rather it is described in the text.
96    PCOC?
278    "Two cautionary notes before we continue:" seems too oddly self-referential.
314    "it converted to a direct, DIDS-sensitive, HCO3--uptake alga" is misleading. It didn't convert to another type of alga, It converted to another physiological state or some such.
324    Correct spelling for "depts"
359    CO2 levels "fell" to higher levels? Is this meant to state simply that they were 2-16x higher than present?
363    This is a single sentence paragraph and it's unclear why it is placed here or what its context is. Please integrate elsewhere or provide further justification.
377    Correct to "point" and the following line uses "commonest", which is very informal or abnormal usage (vs "most common").
412    The sentence beginning here is incredibly long and the point becomes rather lost by the end, which is an odd hanging comma-separated statement.
417    Remove "also"
418    Why "workers"?
468    Futue research couldn't "bring about" interactions. Perhaps correct to "illuminate" or something of that sort.
480    Delete superfluous "and"

Comments on the Quality of English Language

All comments in previous section.

Reviewer 2 Report

Comments and Suggestions for Authors

This is a well- written paper in which the authors summarize the current understanding of photosynthetic carbon assimilation by submerged marine photoautotrophs.

I have no major requests or comments, just some suggestions

Table 1: Can you list the references next to the values taken from them?
Lines 153-157: Is the range of 30 and 85 μM documented for all of these listed species or does each species have its own specific value? It would be nice to list them all in Table like Table 1 and provide references next to each species.

Reviewer 3 Report

Comments and Suggestions for Authors

This article presents a review of the article “Inorganic Carbon Acquisition and Photosynthetic Metabolism in Marine Photoautotrophs: A Summary.” The authors address the relevant topic of inorganic carbon uptake by marine photoautotrophs, which is of great importance for understanding biogeochemical cycles and the productivity of marine ecosystems. However, despite interesting research directions, the text contains several shortcomings, the correction of which would improve its perception by readers.

It is recommended to expand the analysis of differences among taxonomic groups of marine photosynthetic organisms and to pay more attention to specific mechanisms characteristic of cyanobacteria, diatoms, and green algae, with an emphasis on the localization and functional features of different carbonic anhydrase isoforms, whether in the periplasm, chloroplasts, or carboxysomes. This would allow for a deeper understanding of the variability of photosynthetic strategies and the evolutionary adaptation of these organisms.

Another important revision is a more detailed examination of carbon-concentrating mechanisms (CCM). It is advisable to conduct a comparative analysis of the primary transport systems, such as Na⁺-dependent HCO₃⁻ permeases, SLC4 antiporters, and NDH-1 complexes, as well as to consider the interactions of carboxysomes in different cyanobacterial species. Such an analysis would help substantiate mechanistic differences and provide a more objective assessment of the role of each mechanism in photosynthetic processes (https://doi.org/10.3389/fmicb.2022.933160).

Furthermore, it is important to focus on the differences between macro- and microalgae. It would be useful to include in the analysis data on HCO₃⁻ buffering in brown algae such as Macrocystis pyrifera (Subiabre, P. A. F. (2015). The effects of ocean acidification on photosynthesis, growth, and carbon and nitrogen metabolism of Macrocystis pyrifera), as well as to examine the role of extracellular carbonic anhydrase in species such as Posidonia oceanica (https://doi.org/10.1016/S0022-0981(98)00172-5). Clarifying these aspects would allow for a more comprehensive understanding of the diversity of photosynthetic strategies depending on the morphological and physiological characteristics of organisms.

Despite mentioning photorespiration, the article would significantly benefit if it included a more detailed analysis of its expression in different groups. The authors are recommended to further study the impact of decreased pH, associated with ocean acidification, on photorespiration processes, which may contribute to increased productivity in some ecosystems. Such an analysis would not only explain current observations but also predict which species may benefit or suffer under climate change conditions.

It is also necessary to reassess the validity of the study’s conclusions. The statement that most marine plants prefer to use HCO₃⁻ requires additional evidence, especially considering that red algae of the genera Porphyra and Gelidium predominantly utilize CO₂ (DOI:10.1016/j.biortech.2019.121700). Similarly, it would be useful to introduce a clearer classification and assessment of the efficiency of the Rubisco enzyme among different groups, which would allow for a better understanding of the features of their photosynthetic strategies.

Comments on the Quality of English Language

The English could be improved to more clearly express the research.

Reviewer 4 Report

Comments and Suggestions for Authors

Recommendation: Major Revision

Title: Inorganic Carbon Acquisition and Photosynthetic Metabolism in Marine Photoautotrophs: A Summary

General Comments:

The review paper is informative and provides a good review of the  inorganic carbon acquisition and photosynthetic metabolism in marine photoautotrophs. While the topic is engaging, it largely revisits well-established research. Certain sections of the paper lack adequate detail (needs additional support form recent reference papers), and important points are sometimes missing or insufficiently explained. The review comments need to be addressed before the manuscript can be considered for publication.

Review Comments:

  • The authors mentioned "the dogma has been that net photosynthesis... supports growth down to a depth where 1% of surface irradiance remains” but it lacks citations beyond older references. Are there more recent studies supporting or challenging this statement? In addition, the statement about marine photosynthesis adaptations should be supported by updated research done from the past five years.
  • In section 2, the authors discussed marine inorganic carbon sources and introduces Equation 1. I suggest that the authors consider introducing the equation with a statement on why carbonate chemistry is fundamental to marine photosynthesis. The text jumps between numerical data, chemical equations, and biological implications without smooth transitions. I suggest that the authors revised this portion to maintain a smooth thread of discussion.
  • In section 3, the authors discussed Rubisco including several equations and kinetic parameters. I suggest that they provide an overview of the importance of these values matter in the broader context of terrestrial vs. marine photosynthesis. A summary statement at the end of the section, synthesizing why Rubisco’s kinetic properties impact marine carbon assimilation, would be beneficial.
  • The authors mentioned "If the final pH is around 9 or higher, then this is taken as evidence for HCO3- use since the CO2 concentration is extremely low at the high pH values generated in closed systems." I suggest that the authors further elucidate the logic behind using pH-drift experiments as definitive proof of HCO3- utilization. In addition, the limitations and potential confounding factors (such as species-specific responses, buffering effects or ion exchange) should be also discussed.
  • The authors used Ulva as a model organisms for CCM characteristics. How about other representative species of macroalgae. I suggest that they provide additional discussion of the differences between taxonomic groups. Is it possible to provide examples from other groups such as red or brown seaweeds?
  • I suggest that the authors provide quantitative comparisons to improve the discussion of pH manipulation experiments. Is it possible that actual data ranges from previous studies be provided to improve the scientific argument in this portion of the paper?
  • I noticed that a large section of the review paper focuses on Ulva And the authors sometimes make generalizations about macroalgae and seagrasses in terms of carbon acquisition strategies. I suggest that the authors expand this discussion part by adding other algal lineages to improve scientific soundness.
  • The authors mentioned that CCM-dependent species may not show improved performance under elevated CO2 but lacks an explanation. I suggest that deeper analysis be done as to how energetic trade-offs influence growth as well as whether downregulation of CCMs offsets potential CO2 fertilization effects can improve this portion.
  • In the future scenario section, the authors speculates on potential physiological changes without discussing ecological or evolutionary adaptation. I believe that genetic adaptation and acclimation to changing CO2 conditions should be considered, as well as the possibilities of shifts in species composition favoring those with flexible carbon acquisition strategies in the environment.

Round 2

Reviewer 3 Report

Comments and Suggestions for Authors

The manuscript has been revised taking into account the review, and most of the comments have been answered.